# Noncanonical binding of BiP ATPase domain to Ire1 and Perk is dissociated by unfolded protein $C_H1$ to initiate ER stress signaling

Marta Carrara[†], Filippo Prischi, Piotr R Nowak, Megan C Kopp, Maruf MU Ali*

Department of Life Sciences, Imperial College, London, United Kingdom

**Abstract** The unfolded protein response (UPR) is an essential cell signaling system that detects the accumulation of misfolded proteins within the endoplasmic reticulum (ER) and initiates a cellular response in order to maintain homeostasis. How cells detect the accumulation of misfolded proteins remains unclear. In this study, we identify a noncanonical interaction between the ATPase domain of the ER chaperone BiP and the luminal domains of the UPR sensors Ire1 and Perk that dissociates when authentic ER unfolded protein $C_H1$ binds to the canonical substrate binding domain of BiP. Unlike the interaction between chaperone and substrates, we found that the interaction between BiP and UPR sensors was unaffected by nucleotides. Thus, we discover that BiP is dual functional UPR sensor, sensing unfolded proteins by canonical binding to substrates and transducing this event to noncanonical, signaling interaction to Ire1 and Perk. Our observations implicate BiP as the key component for detecting ER stress and suggest an allosteric mechanism for UPR induction.

*For correspondence: maruf.ali@
imperial.ac.uk

Present address: [†]MRC
Laboratory of Molecular Biology,
University of Cambridge,
Cambridge, United Kingdom

**Competing interests:** The
authors declare that no
competing interests exist.

**Reviewing editor**: Stephen C
Harrison, Howard Hughes
Medical Institute, Harvard
Medical School, United States

## Introduction

The endoplasmic reticulum (ER) is an essential eukaryotic organelle responsible for a number of processes including folding and maturation of secretory proteins destined for the extracellular space. The sudden requirement for processing large quantities of secretory proteins can be immense and results in over burdening the folding machinery within the ER, leading to accumulation of misfolded proteins and ER stress (*Malhotra and Kaufman, 2007*; *Walter and Ron, 2011*; *Wang and Kaufman, 2012*). The unfolded protein response (UPR) is a cell signaling system that detects the presence of misfolded proteins within the ER and carries out a varied cellular response to maintain homeostasis (*Malhotra and Kaufman, 2007*; *Hetz et al., 2011*; *Walter and Ron, 2011*; *Wang and Kaufman, 2012*). Both Ire1 and Perk are UPR sensor proteins possessing luminal domains that are involved in detecting the presence of unfolded protein, although the precise mechanism is unclear (*Malhotra and Kaufman, 2007*; *Hetz et al., 2011*; *Wang and Kaufman, 2012*; *Carrara et al., 2013*). Early studies within the field provide evidence for the role of BiP (ER Hsp70 chaperone) in UPR activation by binding to the luminal domains and maintaining them in an inactive state (*Bertolotti et al., 2000*; *Liu et al., 2000*; *Okamura et al., 2000*; *Ma et al., 2002*; *Zhou et al., 2006*). An alternative model proposed a direct recognition of misfolded proteins by Ire1 (*Credle et al., 2005*; *Gardner and Walter, 2011*; *Promlek et al., 2011*).

In this study, we set out to glean new insights into the mechanism of mammalian UPR activation, by initially reconstituting the mechanistic events in vitro, using recombinant human Ire1 and Perk luminal domain proteins, in the presence of authentic and relevant ER unfolded protein $C_H1$; and assessing whether BiP is involved in this process. By primarily using biophysical/biochemical techniques, we discover a direct noncanonical interaction between the ATPase domain of BiP to the luminal domains of Ire1 and Perk, clearly indicating a UPR signaling role. This interaction is unaffected by nucleotide

**eLife digest** Proteins perform many essential tasks in cells, but to be able to work they first have to correctly fold into a specific three-dimensional shape. Within the cell, many proteins are folded with the help of 'chaperone' proteins. If any proteins fold incorrectly, the normal workings of the cell can be disturbed, which may damage the cell. This is more likely to happen if a cell suddenly requires a large number of proteins to be made, which can overwhelm the chaperone proteins.

In humans and other eukaryotic organisms, many proteins are folded in a compartment within the cell called the endoplasmic reticulum. Inside this compartment there is a system called the unfolded protein response that detects misfolded proteins and boosts the cell's capacity to re-fold them. As part of this system, two sensor proteins detect when misfolded proteins are present, but it is not clear how they do so. It has been suggested that a chaperone protein called BiP may be able to activate these sensor proteins in order to turn on the unfolded protein response.

In this study, Carrara et al. studied the sensor proteins and BiP using an artificial set-up in the laboratory. The experiments show that both of the sensor proteins can bind to a section of the BiP chaperone called the ATPase domain. However, in the presence of an unfolded protein, BiP stopped interacting with the sensor proteins, which could allow the sensor proteins to activate the unfolded protein response. The experiments also show that BiP must bind to the unfolded protein to activate the unfolded protein response.

Carrara et al.'s findings suggest that BiP has a dual role in cells: to sense unfolded proteins by binding to them, and then to activate the sensor proteins that trigger the unfolded protein response.

Together, these results suggest a new model for how cells detect and respond to misfolded proteins within the endoplasmic reticulum, and may provide new targets for therapies to treat diseases caused by defects in protein folding.

binding to BiP. We further show that unfolded protein $C_H1$ binds to the canonical BiP substrate binding domain; this relieves the interaction between BiP and luminal domains of Ire1 and Perk. Moreover, our data indicate that this model is consistent in cells. Overall, our observations suggest a novel allosteric model for UPR induction that involves BiP as the key component for detecting ER stress.

## Results

### Noncanonical binding of Ire1 and Perk luminal domains to BiP ATPase domain

We initially expressed and purified full-length human Ire1 and Perk luminal domains along with full-length BiP encompassing both the ATPase domain and the substrate binding domains. To assess the role of BiP in UPR sensing, we first examined whether there was an interaction between the luminal domains of Ire1 and Perk to full-length BiP. Using microscale thermophoresis, we found that both Ire1 and Perk luminal domains bound to BiP with binding affinities of $K_d = 1.33$ μM and 1.92 μM, respectively, consistent with a typical transient protein–protein interaction (*Figure 1A–B*). To dissect the molecular basis of the interaction between Ire1 and Perk with BiP, we tested if this interaction was mediated by the canonical substrate binding domain, as it is the case for this chaperone, or the ATPase domain. Both BiP's ATPase and substrate binding domains were expressed and purified separately and assessed for their ability to bind to Ire1 and Perk luminal domains. Analysis of binding by thermophoresis revealed that in contrast to full-length BiP, we measured no binding between luminal domains and BiP's substrate binding domain (*Figure 1C–D*). Surprisingly, we discovered that BiP's ATPase domain bound to both Ire1 and Perk luminal domains with binding affinities of $K_d = 1.97$ μM and 2.05 μM, which were almost identical to full-length BiP (*Figure 1C–D*). These results reveal that BiP binding to the luminal domains of Ire1 or Perk does not involve BiP's substrate binding domain. Rather, the interaction between BiP with Ire1 and Perk luminal domains is entirely mediated by BiP's ATPase domain (*Figure 1C–D*). Such an interaction with BiP ATPase domain is novel, and suggests that the BiPATPase domain interaction with Ire1 or Perk is distinct from the classical chaperone-substrate interaction, and may serve some key signaling functions that we set out to elucidate. To confirm that the noncanonical interactions detected by microscale thermophoresis were robust, we developed

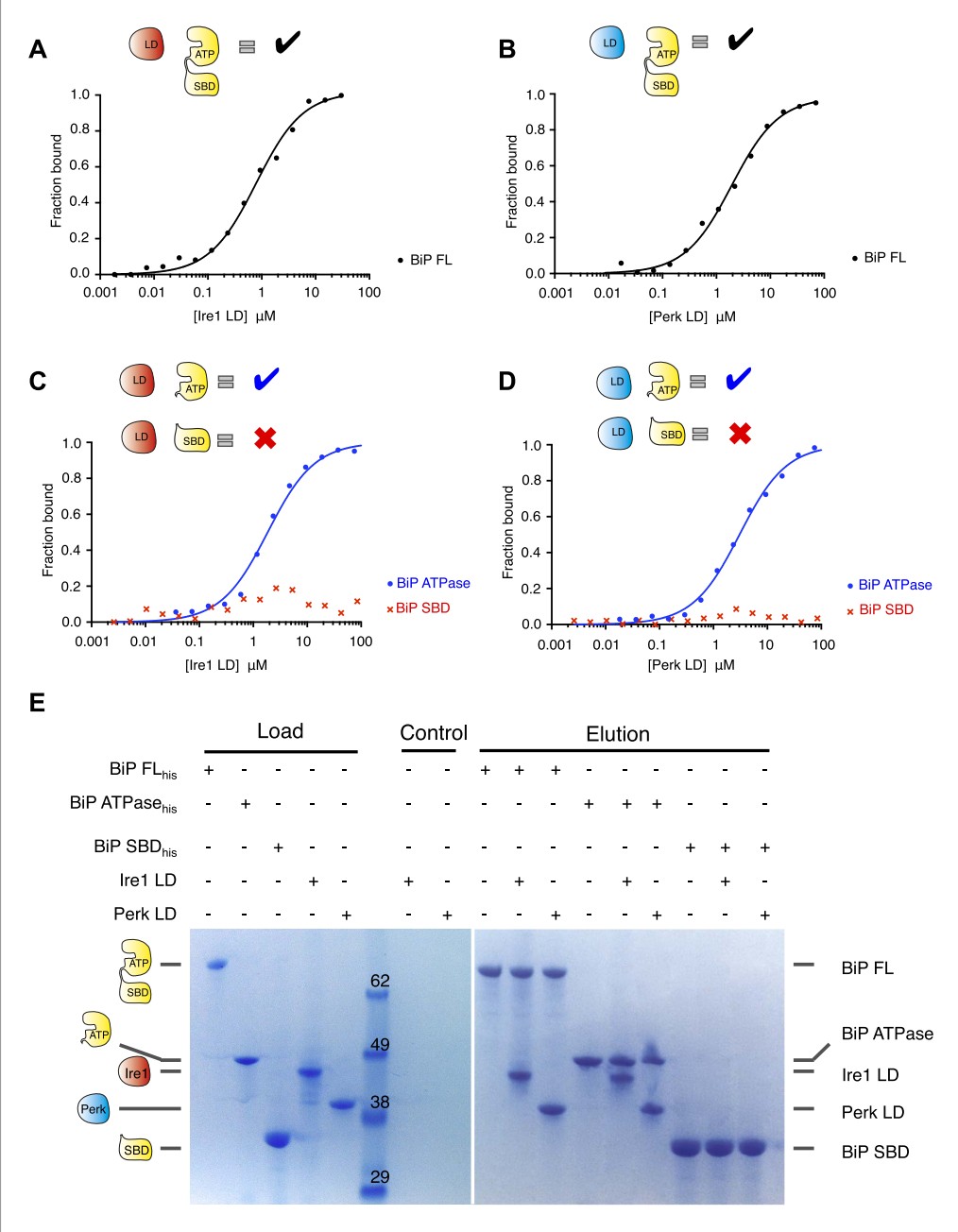

**Figure 1**. Noncanonical binding of BiP ATPase domain to Ire1 and Perk. (**A–B**) Microscale thermophoresis (MST) analysis showing sigmoidal binding curves for interaction between full-length BiP and the complete luminal domains (region I–V) of (**A**) Ire1 luminal domain ($K_d$ = 1.33 µM) and (**B**) Perk luminal domain ($K_d$ = 1.92 µM). (**C–D**) MST binding curves of interaction between BiP sub-domains (ATPase and substrate binding domain) and (**C**) Ire1 luminal domain (ATPase $K_d$ = 1.97 µM; no binding to substrate binding domain) and (**D**) Perk luminal domain (ATPase $K_d$ = 2.05 µM; no binding to substrate binding domain). (**E**) Pull down assay showing BiP-luminal domain complexes using His$_6$-tagged BiP proteins and luminal domains of Perk and Ire1 visualized by coomassie brilliant blue stained SDS PAGE gel. Ire1 and Perk luminal domains bind to full-length BiP and BiP ATPase domain. No binding to BiP substrate binding domain was observed.

an independent assay. Pull down assays with purified proteins recapitulated our previous results: His$_6$-tagged full-length BiP and BiP ATPase domain proteins were able to form an interaction with luminal domain of Ire1 and Perk, while His$_6$-tagged BiP substrate binding domain did not bind to Ire1 and Perk luminal domains (*Figure 1E*). Thus, we have here recapitulated the interaction

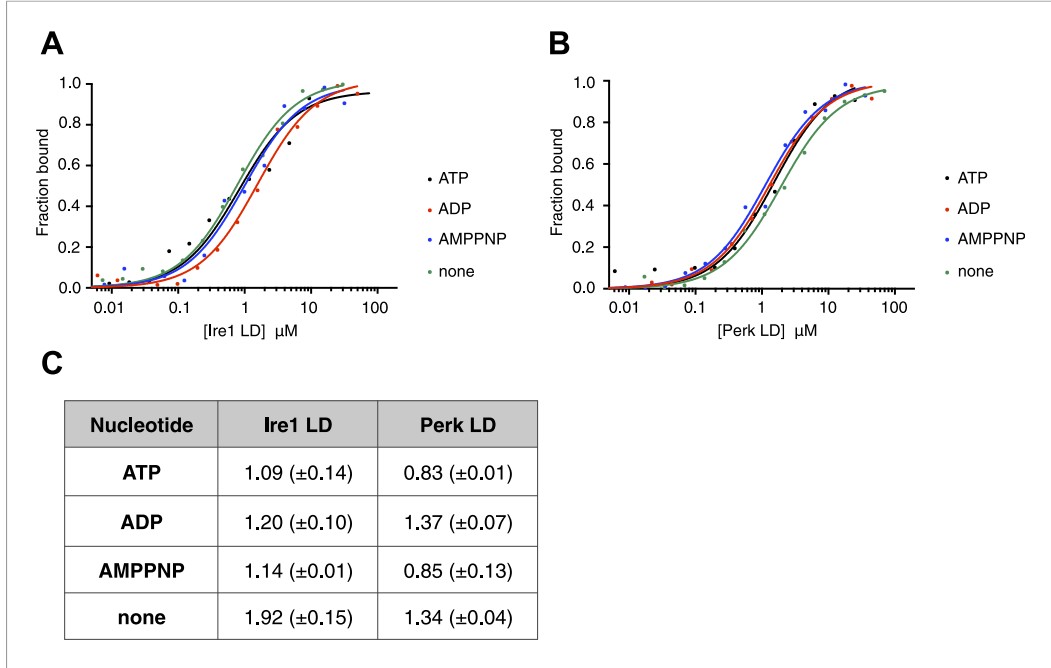

**Figure 2**. The noncanonical binding of BiP ATPase domain to Ire1 and Perk is independent of nucleotides. (**A–B**) MST analysis showing sigmoidal binding curves for full-length BiP interaction with (**A**) Ire1 luminal domain and (**B**) Perk luminal domain in the presence of 10 mM ATP, ADP, AMPPNP and in the absence of nucleotides. (**C**) List of $K_d$ values ($\mu$M ± SE) for Ire1 and Perk luminal domain interactions with full-length BiP in the presence of nucleotides for the binding curves represented in **A** and **B**. Binding between luminal domains and BiP was not affected by the presence of the various nucleotides.

between full-length BiP and Ire1 and Perk luminal domains. Moreover, we confirm that Ire1 and Perk luminal domains bind to BiP's ATPase domain.

## Noncanonical binding of BiP ATPase to Ire1 and Perk is independent of nucleotides

ATP–ADP cycling is an important part of BiP's chaperone activity (*Mayer et al., 2003*). To further characterize the novel interaction between BiP, Ire1, and Perk, we next assessed if the formation of the complex between BiP ATPase domain and the UPR luminal domain was affected by the presence of nucleotides. Using thermophoresis, we measured the binding of full-length BiP to luminal domains of Ire1 and Perk in the presence of 10 mM ATP, ADP, AMPPNP, and also in the absence of nucleotide. We observed the affinity of interaction was very similar both in the presence of the various nucleotides, and when nucleotide was absent. Therefore, the addition of ATP, AMPPNP, and ADP had no effect upon the formation of the full-length BiP-Perk and full-length BiP-Ire1 luminal domain complexes (*Figure 2A–C*). This indicates that the interaction between BiP ATPase domain and Ire1 or Perk is unrelated to the interaction between BiP and its canonical substrates.

## BiP binds to region II–IV of luminal domains

The luminal domain of yeast Ire1 has been divided into five subregions based upon a series of deletion mutant's ability to activate UPR (*Kimata et al., 2004*, *2007*). We designed a range of constructs based upon this classification in order to determine more precisely the interaction site between Ire1 and Perk luminal domains with BiP's ATPase domain (*Table 1*). One point to note, is that using this assignment (*Kimata et al., 2004*, *2007*) yeast Ire1 luminal domain possesses an extended region I, whilst the equivalent region in human Ire1 is essentially absent. The implication for human Ire1 is that both regions I and II are very close together and map onto the equivalent of yeast region II. Next, we measured the binding affinities for BiP's ATPase domain interacting with human Ire1 and Perk luminal domains comprising of various regions between I and V. We found that binding affinities

**Table 1.** Construct sizes for all BiP, Ire1 and Perk *in vitro* constructs used in this study

| Protein | Residue range |
| --- | --- |
| BiP FL | 28–654 |
| BiP ATPase | 28–405 |
| BiP SBD | 422–654 |
| Ire1 I–V | 24–440 |
| Ire1 I–IV | 24–390 |
| Ire1 II–V | 32–440 |
| Ire1 II–IV | 32–390 |
| Ire1 LD for cross-link experiment | 32–390 |
| Perk I–V | 54–509 |
| Perk I–IV | 54–403 |
| Perk II–V | 105–509 |
| Perk II–IV | 54–403 |

for region II–IV were essentially the same as those measured for binding to the full-length luminal domain construct region I–V, indicating the core interaction between UPR luminal domains and BiP maps to the luminal domain region II–IV and ATPase domain, respectively (*Figure 3A–C*). To reinforce the in vitro data, we assessed whether the interaction between Ire1 luminal domain regions II–IV and BiP occurs in cells. Principally, we conducted a co-immunoprecipitation experiment using an Ire1 region V deletion mutant (Ire1ΔV; Δ390–430) with a BiP construct possessing an N-terminal HA-tag, in the absence and presence of ER stress, and then compared with full-length Ire1 (*Figure 3D*). In the absence of ER stress, immunoblotting with Ire1-specific antibody clearly indicates the presence of bands that are consistent for both Ire1 ΔV mutant and full-length protein co-immunoprecipitating with BiP. Upon ER stress, both Ire1 ΔV mutant and full-length Ire1 protein display a reduced level of interaction with BiP (*Figure 3D*). We also assessed direct binding with BiP ATPase domain to both Ire1 full-length and ΔV mutant—observing an interaction consistent with in vitro data (*Figure 3E*). These results suggest that BiP interacts with Ire1 region II–IV in cells and that region V is dispensable for this interaction to occur, reinforcing the previous in vitro analysis (*Figure 3D,E*). Ire1 luminal domain region II–IV has been suggested to be important for stress sensing in yeast (*Credle et al., 2005*). Here, we identified region II–IV as the binding site for human BiP-luminal domain protein interaction. This suggests that BiP is likely to be important in mammalian ER stress sensing.

## Unfolded peptide mimic has no effect upon BiP-luminal domain interactions

The identification of a noncanonical interaction between BiP's ATPase domain and the luminal domains of Ire1 and Perk raises the possibility that this interaction might have a signaling function in initiating UPR activation. In order to probe the significance of this interaction in UPR signaling, we set out to reconstitute ER stress in our system. Therefore, we critically assessed the effect of unfolded peptide substrates, and hence ER stress, upon the BiP-luminal domain complexes using our assays. First, we assessed the previously described yeast Ire1-specific unfolded peptide mimic, ΔEspP (*Gardner and Walter, 2011*). ΔEspP has been shown to bind directly to yeast Ire1 luminal domain and to subsequently cause luminal domain oligomerisation (*Gardner and Walter, 2011*). Although, we measured weak binding between ΔEspP and human luminal domain proteins of Ire1 and Perk (*Figure 4—figure supplement 1A–C*), surprisingly we detected no effect upon full-length BiP-Ire1 and full-length BiP-Perk luminal domain complexes upon addition of peptide (*Figure 4—figure supplement 1D–F*).

## Unfolded proteins bind to canonical BiP substrate binding domain only

To reconstitute ER stress in vitro, we next turned to an authentic and relevant ER-resident unfolded protein to assess the effects of misfolded proteins on the BiP-Ire1 and BiP-Perk complexes. The intrinsically unfolded immunoglobulin constant heavy chain domain ($C_H1$), which is disordered in the absence of its cognate binding partner $C_L$, is a relevant, ER localized unfolded protein substrate (*Feige et al., 2009*; *Marcinowski et al., 2011*). First, we examined if unfolded protein $C_H1$ binds to the luminal domains of Ire1 and Perk. Surprisingly, we measured no interaction between $C_H1$ and luminal domains suggesting that Ire1 and Perk luminal domains are not directly involved in detecting unfolded proteins (*Figure 4A–B*). Next, we assessed the interaction of $C_H1$ to full-length BiP. As expected (*Marcinowski et al., 2011*), we found that $C_H1$ bound to full-length BiP with a binding affinity of $K_d = 8.7$ μM. To identify what region of BiP, $C_H1$ specifically bound to, we used ATPase and substrate binding domain proteins for interaction analysis. As expected for an Hsp70 chaperone (*Marcinowski et al., 2011*), we found that $C_H1$ bound only to the substrate

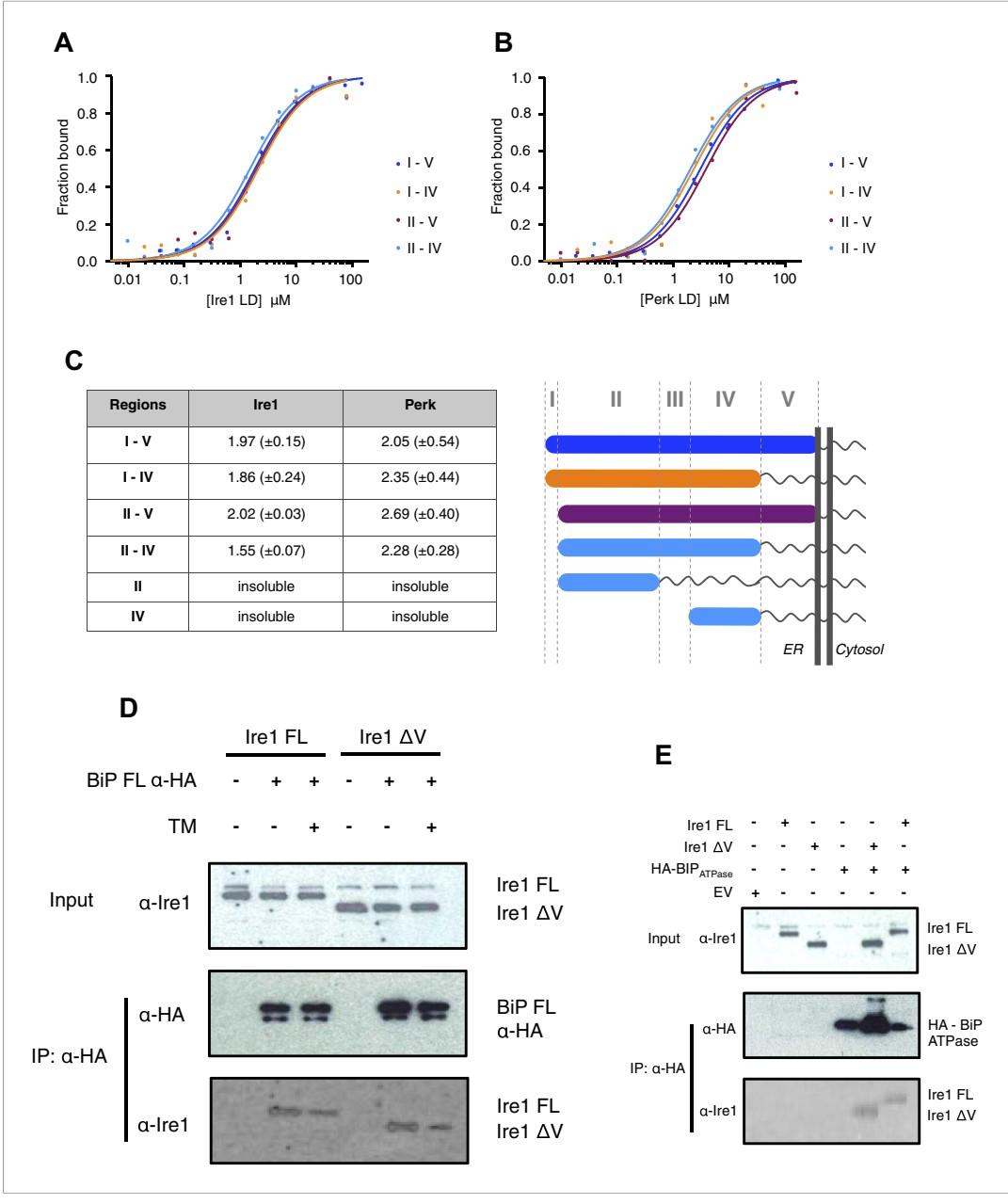

Figure 3. Core interaction between BiP ATPase and luminal domains occurs via region II-IV of luminal domains. (A–B) MST binding curves of interaction between BiP ATPase and different length constructs of (A) Ire1 luminal domain and (B) Perk luminal domain. (C) List of $K_d$ values ($\mu M \pm SE$) for BiP ATPase interaction with the various Ire1 and Perk luminal domain constructs (based on regions I–V) for binding curves represented in A and B. The luminal domain region II–IV, is solely responsible for binding to BiP proteins and regions I and V are dispensable in this interaction. (D) Co-immunoprecipitation experiment in which HEK293T cells were co transfected with either Ire1 mutant lacking region V (Ire1ΔV; Δ390–430) or full-length Ire1, along with HA-tagged BiP, in the absence or presence of ER stress (TM = 5 μM; 4 hr tunicamycin). Immunoprecipitating with HA peptide and then immunoblotting with Ire1 specific antibody reveals an interaction between BiP and both full-length and mutant Ire1 that is missing region V (Ire1 ΔV), which is reduced after ER stress. This interaction in cells reinforces the in vitro data. (E) Co-immunoprecipitation experiment similar to (D), but cells were co transfected with HA-tagged BiP ATPase and were not subjected to ER stress with tunicamycin.

binding domain of BiP with a similar binding affinity to that of full-length BiP, suggesting that BiP's substrate binding domain was solely responsible for binding to $C_H1$ unfolded protein.

## Unfolded proteins dissociate noncanonical BiP-luminal domain interaction

Having established that $C_H1$ binds exclusively to BiP's substrate binding domain and that BiP's ATPase domain is responsible for its interaction with the luminal domains; we evaluated the effects of $C_H1$ binding to the BiP-luminal domain complex using our pull down assay. The addition of $C_H1$ to the complex caused

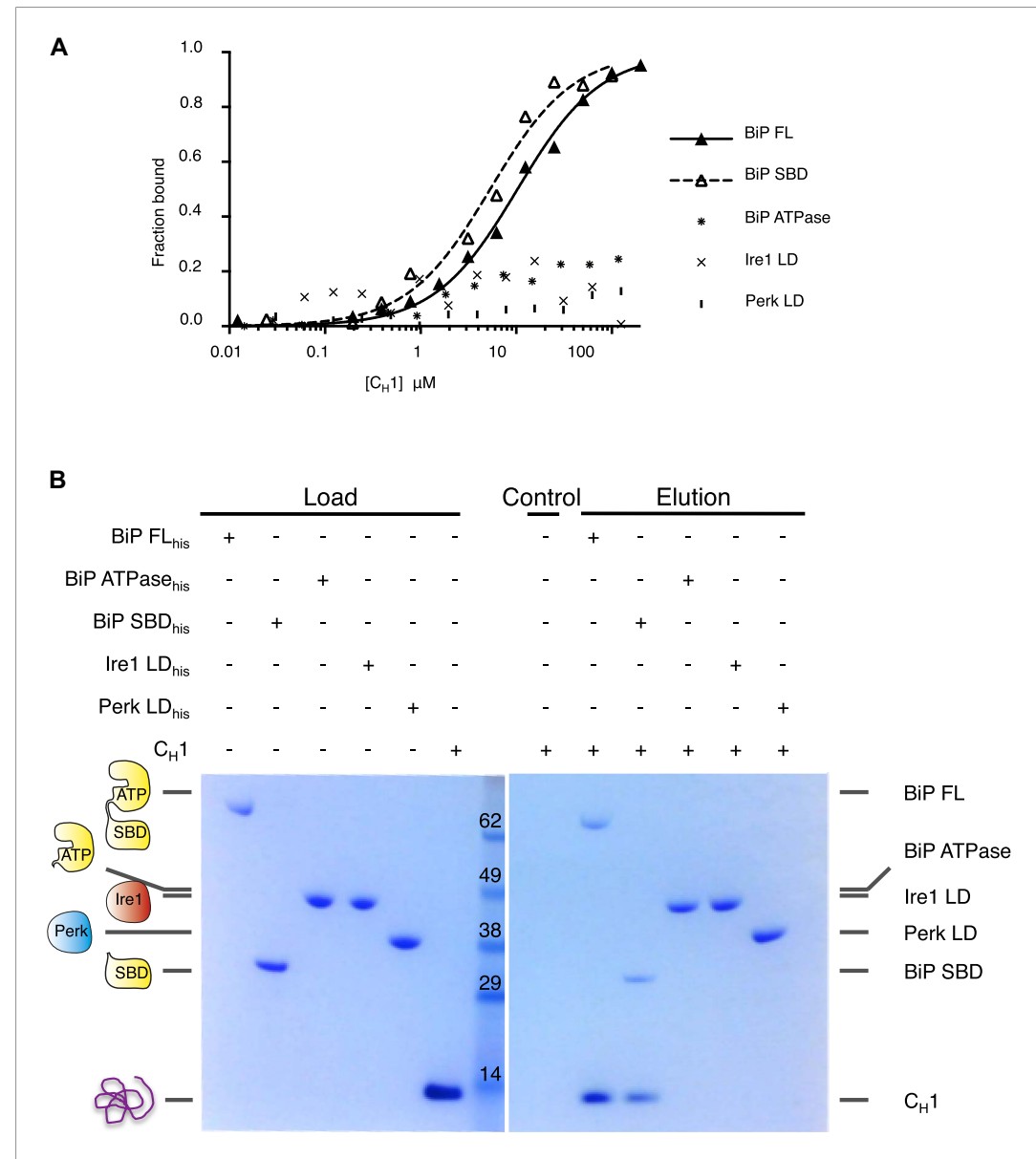

**Figure 4**. Unfolded protein $C_H1$ binds to canonical BiP substrate binding domain without binding to UPR luminal domains. (**A**) MST binding curves for $C_H1$ binding to full-length BiP ($K_d = 8.7$ µM), BiP's ATPase domain (no binding), BiP's substrate binding domain ($K_d = 5.1$ µM), Ire1 luminal domain (no binding) and Perk luminal domain (no binding). (**B**) Pull down experiment showing $C_H1$ binding to both full-length and substrate binding domain of BiP only, with no interaction observed to luminal domains, reaffirming the data for $C_H1$ interactions using MST in part **A**.

The following figure supplement is available for figure 4:

**Figure supplement 1**. Assessing the role of unfolded protein peptide mimic (ΔEspP) in UPR stress sensing.

complete dissociation of both full-length BiP-Ire1 and full-length BiP-Perk luminal domain complexes (*Figure 5A*). The dissociation of the BiP-Ire1 and BiP-Perk complexes by $C_H1$ reveals that unfolded protein binding to BiP substrate binding domain, causes BiP ATPase domain to dissociate from BiP-luminal domain complex. Thus, when BiP is engaged in a signaling complex with Ire1 or Perk, its substrate binding domain remains available to interact directly with misfolded protein. This interaction dissociates the complex and initiates UPR signaling. To confirm these findings, we incubated $C_H1$ with BiP and then subsequently added Ire1 and Perk luminal domains. Consistent with our prediction, we found that BiP engaged with misfolded protein $C_H1$, is unable to bind to the luminal domains of Ire1 or Perk (*Figure 5B*). Thus, the binding of $C_H1$ and luminal domains to BiP are mutually exclusive (*Figure 5B*).

## BiP deletion mutants that lack substrate binding domain attenuate UPR signaling

Thus far, the in vitro data suggest an allosteric/conformational change engendered by unfolded protein binding to BiP's substrate binding domain, causing dissociation of BiP, via it's ATPase domain,

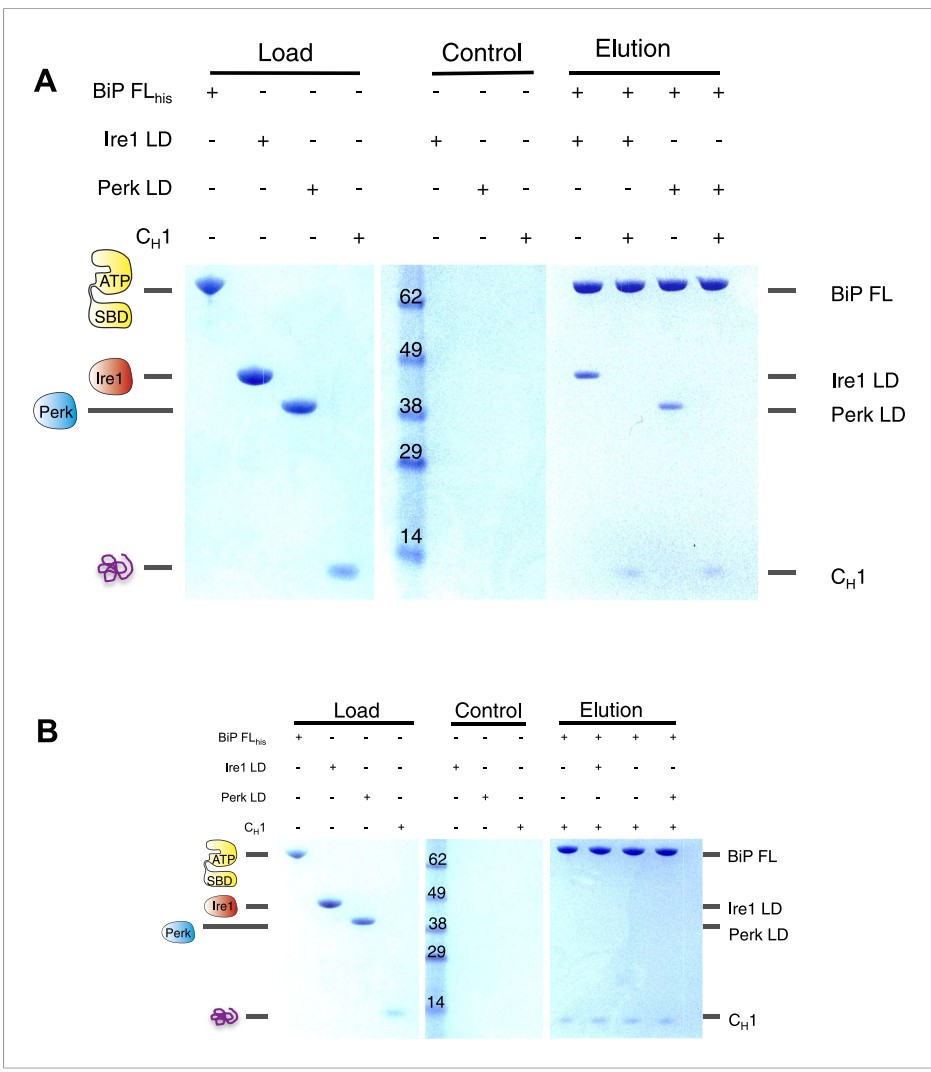

**Figure 5**. The unfolded protein $C_H1$ dissociates the noncanonical interaction between BiP ATPase domain and the luminal domain of Ire1 or Perk. (**A**) Pull down assay assessing the effects upon addition of unfolded protein $C_H1$ to His$_6$-tagged full-length BiP-luminal domain complexes. $C_H1$ disrupts BiP-luminal domain interaction and causes the complexes to dissociate. (**B**) When His$_6$-tagged full-length BiP is initially incubated with $C_H1$ and then subsequently Ire1 and Perk luminal domains are added, we see no binding between luminal domains and BiP indicating that luminal domains and $C_H1$ binding to BiP are mutually exclusive.

from Ire1 and Perk. To test if this model occurs in cells, we co transfected Ire1 and Perk, with HA-tagged full-length BiP, and BiP deletion mutant lacking the substrate binding domain; encompassing BiP ATPase domain only, consistent with earlier in vitro constructs used. We compared the levels of phosphorylated Ire1 and Perk, as indicators of UPR signaling, when expressed with full-length BiP or BiP ATPase domain, in both unstressed and ER stressed cells.

We observed that in ER stressed cells expressing full-length BiP with Ire1 or Perk, exhibited significantly greater levels of phosphorylation when compared to cells expressing BiP ATPase domain with Ire1 or Perk (*Figure 6A,B*). In agreement with our prediction, cells that lacked the substrate binding domain of BiP were unable to efficiently respond to ER stress and displayed significantly less phosphorylation—a result of attenuated UPR signaling. These data suggest that allosteric regulation occurs in cells, consistent with our in vitro model.

## BiP impedes Ire1 luminal domain dimer and tetramer formation

The dissociation of BiP from Ire1 and Perk luminal domains upon ER stress leads to further downstream signaling (*Bertolotti et al., 2000*; *Liu et al., 2000*; *Okamura et al., 2000*). It is highly likely that some sort of oligomerisation event occurs (*Walter and Ron, 2011*), preceding from a dimer state, which makes signal propagation more efficient, although exactly how this happens is not clearly understood. To understand luminal domain oligomerisation and how BiP affects this process, if at all; we analyzed the effects of cross linking upon Ire1 luminal domain protein in the absence and presence of His$_6$-tagged BiP ATPase domain protein. The BiP ATPase domain interacts directly with Ire1 luminal domain, but perhaps more importantly—for clarity of results—it exists as a monomer both in absence and presence of cross linker. Thus, making the identification of the Ire1-BiP multimer bands an easier task. Upon addition of cross linker, Ire1 luminal domain forms two distinct species: a dimer and tetramer state. When we added BiP ATPase in a 1:1 ratio with Ire1 luminal domain protein, and then subjected the mixture to cross linking, we see a reduction in size of the tetramer band; concomitantly, there appears a band that corresponds to a trimer in size (*Figure 7A*). Immunoblotting against His$_6$ peptide reveals that the dimer band in the 1:1 mixture sample contains BiP ATPase; similarly, the trimer band also contains BiP ATPase. The tetramer band is exclusively made up of Ire1 luminal domain protein, and

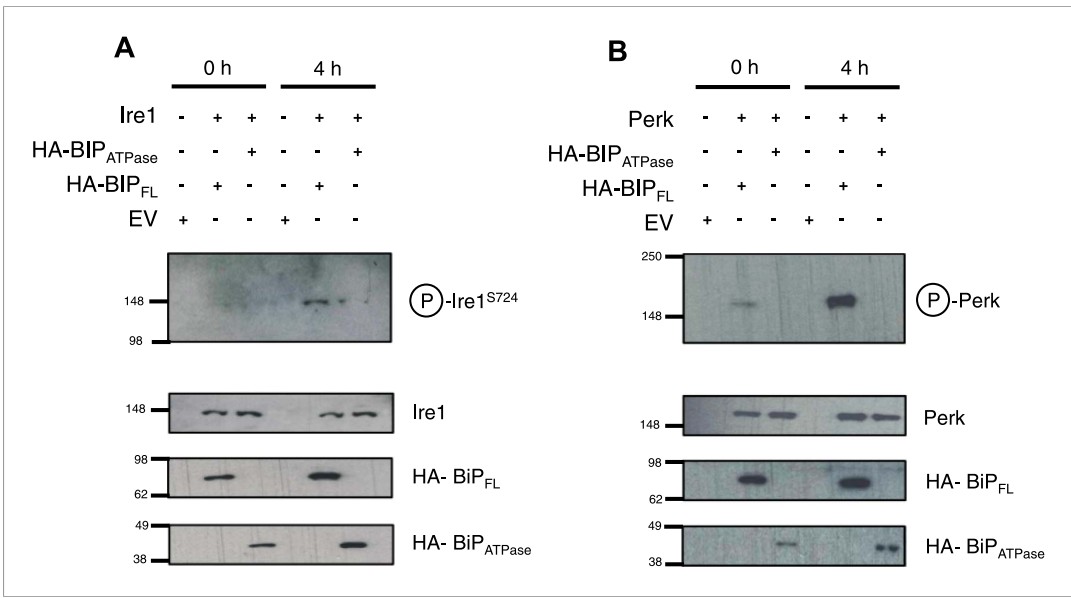

**Figure 6**. BiP deletion mutants, lacking the substrate binding domain, attenuate UPR signaling. (**A**) Ire1 was co expressed with either full-length BiP (HA–BiP$_{FL}$) or BiP ATPase domain, lacking the substrate binding domain (HA–BiP$_{ATPase}$), and challenged with tunicamycin (5 μM) over 0 hr and 4 hr time points in Ire1−/− cells. S724 phosphorylated Ire1 was measured as an indicator of UPR signaling (*Ali et al., 2011*) using pIre1$^{s724}$ antibody (ICR). Cells expressing full-length BiP and Ire1 were fully able to respond to induced ER stress, whilst expression with BiP ATPase domain attenuated UPR signaling. EV = empty vector. (**B**) Similar to (**A**), but expressing Perk full-length in Perk−/− cells and using pPerk antibody (Santa Cruz).

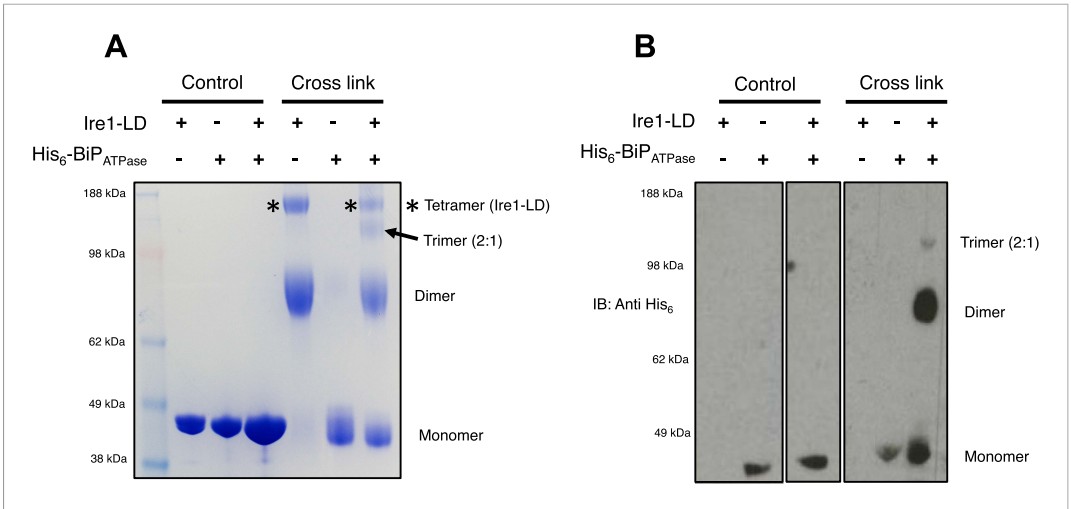

**Figure 7**. BiP impedes Ire1 LD dimer and tetramer formation. (**A**) Ire1 luminal domain (LD; regions II–IV) and His$_6$-tagged BiP ATPase domain proteins, both individually and in 1:1 molar ratio mixture, were visualized as control lanes on a 4–12% Bis-Tris SDS-PAGE gel. The same proteins were then subjected to EGS cross linker for 1 hr, after which the reaction was quenched and samples were visualized along side control lanes. In the presence of cross linker, Ire1 LD forms dimer and tetramer species; when in a mixture with BiP ATPase, there is a reduction in the corresponding tetramer band (*). Also, a band appears that is consistent in size with a trimer species. (**B**) Samples from (**A**) were immunoblotted using anti-His$_6$ antibody, which detects His$_6$-tagged BiP ATPase domain protein. Since BiP ATPase protein is monomeric in the absence or presence of cross linker, BiP ATPase forms hetero dimer and hetero trimer with Ire1 LD. The binding of BiP to Ire1 reduces the size of the tetramer (*) that is exclusively formed by Ire1 LD, leading to the conclusion that BiP inhibits Ire1 LD tetramer formation, by preventing formation of the dimer species.

consequently does not show up when immunoblotting (*Figure 7B*). The data suggest that Ire1 luminal domain dimer and tetramer formation is being impeded by BiP, which binds in a 1:1 hetero dimer and 2:1 hetero trimer interaction, with Ire1. The caveat here is that since full-length BiP may also form dimers, we cannot rule out a 2:2 association between BiP and Ire1. Nonetheless, the data do suggest that BiP may act to impede Ire1 luminal domain dimer and tetramer formation.

## Discussion

In this study, we identify a noncanonical interaction between BiP ATPase and luminal domains of Ire1 and Perk that is independent of nucleotide binding, and hence BiP's chaperone function. Furthermore, we discover that this noncanonical interaction is dissociated by canonical substrate binding to BiP via the substrate binding domain. Thereby, suggesting an allosteric mechanism for UPR induction (*Figure 8*).

There have been several models proposed that describe how misfolded proteins are detected and how this leads to UPR signal activation. An early BiP dependent model describes the central role of BiP in this process (*Bertolotti et al., 2000*; *Liu et al., 2000*; *Okamura et al., 2000*; *Ma et al., 2002*). In this model, the interaction between luminal domains and BiP represses UPR signaling. Upon ER stress, BiP releases from the sensors, Ire1 and Perk, leading to activation. Experimental evidence for this comes from a series of studies that show an interaction between BiP and Ire1 in unstressed cells that dissociate in response to ER stress (*Bertolotti et al., 2000*; *Liu et al., 2000*; *Okamura et al., 2000*; *Ma et al., 2002*; *Oikawa et al., 2009*). However, there have been a number of issues with this model. It was originally thought that this interaction was regulated by nucleotide binding (*Bertolotti et al., 2000*); thereby, suggesting a chaperone substrate type interaction—a non-productive association for UPR signaling—thus complicating analysis. But perhaps more significantly, the model failed to delineate a precise mechanism of BiP release upon accumulation of misfolded proteins, leading to the idea that BiP is competitively titred from luminal domains (*Ron and Walter, 2007*): a notion that has some weakness, not least because UPR is sensitive in response to ER stress.

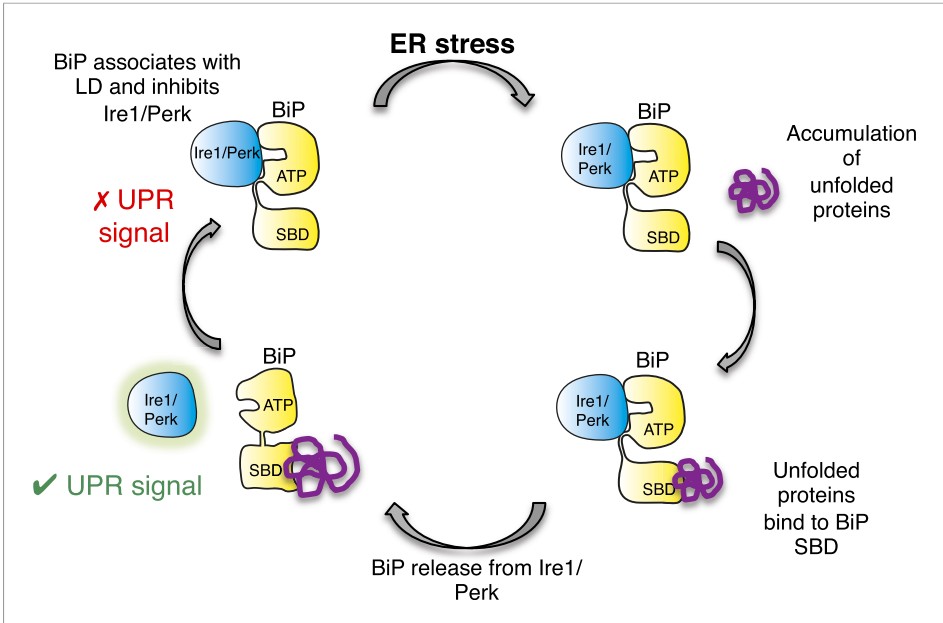

**Figure 8**. Allosteric model of UPR induction. In the absence of misfolded protein, BiP interacts with UPR luminal domains, which acts to repress the UPR signal. Upon ER stress, unfolded protein binds to the canonical BiP substrate binding domain, which in turn causes the noncanonical BiP ATPase-luminal domain interaction to dissociate, ultimately leading to UPR signal activation/propagation.

The present study clarifies this interaction by indicating that it occurs solely via the ATPase domain of BiP, and is unaffected by nucleotides, clearly suggesting a UPR significant role. Previous studies conducted in cells may have been unable to separate this specific ATPase interaction from the chaperone substrate interaction that may occur due to induction of stress and expression of recombinant protein; a process that would no doubt require some BiP acting in a chaperone capacity, and only by analysis in a clean in vitro system that this becomes apparent.

The binding interaction analysis between Ire1 luminal domains (regions I–V) and BiP, to our knowledge is the first biophysical measurements giving specific affinities for association. Surprisingly, we see the previously implicated BiP binding region, Ire1 luminal domain region V, is completely dispensable for association to occur. A key study (*Kimata et al., 2004*) suggested that this region, an area proximal to the ER membrane, was important for BiP binding and that region V deletion mutants were perfectly able to respond to ER stress, leading to the idea that BiP release was not the principal determinant for Ire1 activity—but acts as a first step, followed by unfolded protein binding to luminal domains in a two step mechanism (*Kimata et al., 2004*, *2007*). Structural descriptions of yeast Ire1 luminal domain supported this view (*Credle et al., 2005*). The formation of a groove upon dimerization that resembles an MHC type fold suggested that unfolded proteins bound to luminal domains directly, bypassing BiP for UPR activation. This model has recently gained prominence (*Gardner and Walter, 2011*; *Promlek et al., 2011*); a study showed direct binding of unfolded peptide mimics to yeast Ire1 luminal domain in vitro (*Gardner and Walter, 2011*). The binding of these peptides was observed to cause oligomerisation of luminal domains, with the implication that this leads to UPR activation. In our study, we initially used such peptides to mimic ER stress, observing binding directly to luminal domains, but curiously not to BiP. Moreover, the peptide mimic in our system had no impact upon the noncanonical interaction between BiP ATPase and luminal domains. We noted that certain peptide mimics required addition of a number of charged residues to make it soluble and were derived from non-ER signal peptides. Our attention turned to using $C_H1$ substrate, a previously characterized authentic ER unfolded protein (*Feige et al., 2009*; *Marcinowski et al., 2011*). Using this substrate, we measure robust binding to both full-length BiP and to the substrate binding domain, but not to the ATPase domain of BiP. Furthermore, we do not observe an interaction

between $C_H1$ and luminal domains in vitro. Therefore, the suggestion is that unfolded proteins bind exclusively to BiP's substrate binding domain, a notion that is well accepted for Hsp70 chaperones and for ER isoform BiP (*Marcinowski et al., 2011*, *2013*). Indeed the recent crystal structure of isolated Hsp70 substrate binding domain with unfolded peptide mimics (*Zhang et al., 2014*) indicates that the substrate binding domain alone is also capable to bind unfolded peptide substrates.

Interestingly, a study attempted to reconcile the involvement of BiP in UPR activation with direct binding of unfolded protein to luminal domains (*Pincus et al., 2010*). In this model BiP sequesters inactive Ire1. Upon high levels of ER stress, unfolded proteins bind to Ire1 causing the formation of a higher order active complex, which then recruits inactive monomers from BiP in a competitive fashion. Experimental evidence for this comes from differences in activation between wild-type Ire1 and mutant version of Ire1 (Ire1 ΔV a.k.a Ire1$_{bipless}$) that is unable to bind BiP. The mutant Ire1 was unable to offer any buffering capacity since it was thought to be lacking the BiP binding region, and hence sensitized Ire1 to low unfolded protein load (*Pincus et al., 2010*). The idea that Ire1, BiP, and unfolded proteins exist in some dynamic equilibrium is certainty plausible—however, our data suggest that region V is dispensable for BiP binding to occur. Surprisingly, we do not observe $C_H1$ unfolded protein directly interacting with luminal domains in any of our assays, thus providing evidence against direct association of unfolded protein—a role that is usually reserved for molecular chaperones. Interestingly, one noticeable feature of least some of the previous UPR models mentioned is that they involve a level of competition between components (*Ron and Walter, 2007*; *Pincus et al., 2010*); our present model suggests an allosteric mechanism at the heart of UPR induction.

In summary, we identify a noncanonical interaction between BiP ATPase and luminal domains of Ire1 and Perk that dissociates when unfolded protein binds to the canonical substrate binding domain of BiP. Thus, implicating BiP as a central player in detecting ER stress and suggesting a novel allosteric mechanism for UPR induction.

## Materials and methods

### Expression and purification

All human BiP, Ire1 and Perk proteins used in this study were expressed in *Escherichia coli* BL21 (DE3) cells (Invitrogen, UK) as fusion proteins with an N-terminal His$_6$-tag followed by a PreScission Protease cleavage site. The constructs used are summarized in *Table 1*. All proteins were purified by Co$^{2+}$-NTA affinity using HiTrap TALON crude columns (Clontech, CA) in buffer A (50 mM HEPES (pH 7.5), 200 mM NaCl and 10% glycerol) and eluted in the presence of 250 mM imidazole. Initial lysis and Co$^{2+}$-NTA affinity purifications steps of BiP were supplemented with 5 mM ATP and 10 m MgCl$_2$. Unless otherwise specified, the His$_6$-tag was removed by overnight incubation with PreScission Protease followed by an additional Co$^{2+}$-NTA affinity step to remove any uncleaved protein. Proteins were further purified by anion-exchange using a HiTrap Q HP column (GE Healthcare, UK) and size-exclusion chromatography on a HiLoad 16/60 Superdex 200 column in buffer B (50 mM HEPES [pH 7.5], 75 mM NaCl, 10% glycerol, and 1 mM TCEP). $C_H1$ protein was expressed as previously described (*Marcinowski et al., 2011*). Soluble ΔEspP (MKKHKRILALCFLGLLQSSYSAAKKKK) was purchased from AltaBiosciences (*Gardner and Walter, 2011*).

### Pull down assay

All pull down experiments were carried out in 5 ml gravity flow columns. 50 μl of TALON resin pre-equilibrated with buffer B was incubated with 50 μl of purified BiP$_{his}$ protein at 25 μM for 1 hr at RT. The resin was washed with 1 ml of buffer B to remove any unbound BiP$_{his}$. BiP$_{his}$ was replaced by buffer B in control experiments. Then, 200 μl of purified untagged Ire1 and perk LD or $C_H1$ proteins at 500 μM were added and incubated for 1 hr at RT. The resin was extensively washed with a total of 5 ml of buffer B in 500 μl volumes. For competition pull-downs, 200 μl of Ire1 LD, Perk LD, $C_H1$ or ΔEspP at 500 μM in buffer B were then added, incubated for a further 1 hr at RT and washed as previously with buffer B. Buffer B was supplemented with 10 mM ATP, ADP or AMPPNP plus 10 mM MgCl$_2$, and 30 mM KCl where specified. Finally, the resin was resuspended

with 50 μl of buffer, spun at 10000×g for 5′ and the resulting supernatant was analyzed on a 4–12% gradient SDS-PAGE gel.

## Microscale thermophoresis (MST)

MST experiments were carried out using a Monolith NT.115 instrument (NanoTemper Technologies, Germany). Buffer B was used for all experiments and where specified additional 10 mM ATP, ADP or AMPPNP; and 10 mM MgCl₂, 30 mM KCl were included. Proteins were labeled using the Monolith NT Protein labeling Kit Red-NHS at 50 nM concentration and mixed with equal volumes of sixteen twofold serial dilutions of the unlabeled binding partner. Experiments were carried out in standard treated capillaries with 100% LED power and 80% IR-laser at 25°C. NanoTemper Analysis 1.2.101 software was used to fit the data with a nonlinear solution of the law of mass action and $K_d$ values were determined. Each measurement was repeated in three independent experiments and $K_d$ values were averaged. Standard error (SE) values are shown.

## Cross linking

The homobifunctional protein cross linker ethylene glycolbis(succinimidylsuccinate) (EGS) (Thermo Scientific Pierce, MA) was solubilised in DMSO at a final concentration of 20 mM. BiP, Ire1 and BiP-Ire1 complex were diluted to a final concentration of 50 μM with the reaction buffer (50 mM Hepes pH 8.0, 50 mM NaCl, 5% glycerol and 5 mM DTT). Proteins were incubated with 50-fold molar excess of EGS for 1 hr. The reaction was then quenched for 15 min adding Tris buffer at a final concentration of 50 mM. Samples were first diluted to a final concentration of 10 μM with reaction buffer and then to 5 μM with Laemmli buffer (Sigma). Samples were boiled for 10 min and loaded in NuPAGE 4–12% Bis-Tris pre-cast polyacrylamide gel.

For the western blot, gel was transferred to nitrocellulose membrane (Invitrogen's iBlot) and blocked overnight at 4°C in PBST (PBS in presence of 0.1% Tween 20) + 5% non fat dry milk. Primary anti-His antibody was added to PBST + 2% non fat dry milk in concentration of 1:10000 (Sigma) for 1 hr at room temperature. The membrane was then washed three times in PBST buffer and incubated with anti mouse-HRP antibody. Secondary antibody was diluted (1:10000) in PBST + 2% non fat dry milk and was incubated for 1 hr at room temperature. Followed by another three washes, blots were visualized by Millipore Luminata Crescendo Western HRP substrate and developed on Amersham Hyperfilm ECL.

## Cell culture—Co-immunoprecipitation

Human Embryonic Kidney cell (HEK293T) was cultured in Dulbecco's Modified Eagle Medium supplemented with 10% Fetal Bovine Serum, 2 mM L-Glutamine, 50 U Penicillin/50 μg Streptomycin/ml, 50 μM 2-Mercaptoethanol, and non-essential amino acid ×1. A day before transfection 1,000,000 cells/well (5 ml) were plated on 60 mm tissue culture plates. DNA containing either pcDNA3.1 (empty vector control) or Ire1ΔV, HA-BiP, up to a concentration of 6 μg total, were mixed with Fugene 6 reagent (Promega, WI) in ratio 1:3, and then used to transfect cells. After 48 hr, cells were lysed by 450 μl non-denaturing lysis buffer (+HALT Proteases Inhibitors Coctail, Pierce), scraped and centrifuged. Supernatant was co-immunoprecipitated by anti-HA agarose, mixed with Laemmli buffer, boiled and run on Tris-Glycine 4–12% gel.

Immunoblotting—Gels were transferred to nitrocellulose membrane and blocked in TBST buffer plus 5% Marvel Dried Milk 1 hr in RT. Next, anti-Ire1 (Abcam, UK) and anti-HA were added to blocking buffer (TBST + 1% milk powder) and incubated 1 hr in RT. After that membranes were washed three times in TBST buffer and incubated with secondary antibody in TBST + 2% milk: anti-rabbit (Cell Signaling) for Ire1 and anti mouse for anti-HA, respectively. After 1 hr incubation in RT and another three washes, blots were visualized by Millipore Luminata Crescendo Western HRP substrate and developed on Amersham Hyperfilm ECL.

## Cell culture—Co expression

Ire1−/− and Perk−/− MEF cells (gift from Prof David Ron) were cultured in Dulbecco's Modified Eagle Medium supplemented with 10% Fetal Bovine Serum, 2 mM L-Glutamine, 50 U Penicillin/50 μg

Streptomycin/ml, 50 μM 2-Mercaptoethanol, and Non-Essential Amino Acid ×1. A day before transfection, 500000 cells/well (2 ml) were plated on 6-well plate. DNA containing either pcDNA3.1 (empty vector control) or Ire1, Perk, HA-BiP, HA-BiP ATPase domain, up to a concentration of 3 μg total, were mixed with Fugene HD reagent (Promega) in ratio 1:6, and then used to transfect cells. After 24 hr for Ire1−/−, and 48 hr for Perk−/−, cells were induced by 5 μM tunicamycin dissolved in DMSO (0.5% vol/vol) and harvested after 0 hr and 4 hr. Next, cells were lysed by 250 μl non-denaturing lysis buffer (+HALT Proteases Inhibitors Cocktail, Pierce), scraped and centrifuged. Supernatants were then mixed with 2× Laemmli sample buffer, boiled and run on Tris-Glycine 4–12% gel. For Perk general antibody analysis, cells were immunoprecipitated by Dynabeads Protein G (Life Technologies).

Immunoblotting—gels were transferred to nitrocellulose membrane and blocked in TBST buffer plus 5% Marvel Dried Milk 1 hr in RT. Next, anti-pPerk (Santa Cruz, CA), anti-Perk (Cell Signaling), anti Ire1(abcam), anti p-Ire1$^{s724}$ (Prof Ian Collins, ICR), and anti-HA antibody (Life Technologies) were added to blocking buffer (TBST + 1% milk powder) and incubated 1 hr in RT. After that membranes were washed three times in TBST buffer and incubated with secondary antibody in TBST + 2% milk: anti-rabbit (Cell Signaling) for anti-pPERK and PERK, and anti mouse (GE) for anti-HA. After 1 hr incubation in RT and another three washes, blots were visualized by Millipore Luminata Crescendo Western HRP substrate and developed on Amersham Hyper-film ECL.

## Isothermal Titration Calorimetry (ITC)

Complex formation between ΔEspP and BiP, Ire1 or Perk was measured by ITC using a MicroCal VP-ITC system. ΔEspP interaction with Ire1 or Perk was carried out in buffer B; ΔEspP and BiP interaction was carried out in buffer B plus 10 mM ADP, 10 mM MgCl$_2$, and 30 mM KCl. All experiments were performed at 25°C. The sample cell contained BiP, Ire1 or Perk at approximately 40 μM concentrations and the syringe contained ΔEspP at approximately 450 μM concentrations. Heat of dilution, as determined by titrating ΔEspP into the buffer alone, was subtracted from the raw titration data before analysis. Data were fit by least-squares procedure assuming a one-site binding model using Microcal Origin (version 7.0). K$_d$ values were averaged over three measurements, standard error values are indicated.

## Size-Exclusion Chromatography Multi-Angle Light Scattering (SEC MALS)

To measure the absolute MW of protein species, SEC MALS was carried out using an Agilent 1260 system equipped with a miniDAWN TREOS (Wyatt Technologies) Light Scattering detector and an Optilab T-rEX (Wyatt Technologies) Refractive Index detector. Briefly, 100 μl Ire1 or Perk at 100 μM was mixed with excess ΔEspP (100 μl at 500 μM) for 15′ at RT. In control experiments, 100 μl of buffer B was added instead. Samples were run on a Superdex 200 PC 3.2/30 column (GE Healthcare) pre-equilibrated with buffer B. Data were analyzed with the ASTRA software (Wyatt Technologies, CA).

## Acknowledgements

We thank Magda Reis for the initial cloning, Prof P Freemont and Prof X Zhang and their respective laboratories use of equipment especially for MST. We also acknowledge the Centre for Structural Biology at Imperial College. This work was funded from a Cancer Research UK CD Fellowship (C33269/A11161) and MRC project grant (MR/L007436/1) awarded to MA.

## Additional information

### Funding

| Funder | Grant reference number | Author |
| --- | --- | --- |
| Cancer Research UK | C33269/A11161 | Maruf MU Ali |
| Medical Research Council (MRC) | MR/L007436/1 | Maruf MU Ali |

| Funder | Grant reference number | Author |
|---|---|---|

The funders had no role in study design, data collection and interpretation, or the decision to submit the work for publication.

## Author contributions

MC, FP, PRN, Conception and design, Acquisition of data, Analysis and interpretation of data; MCK, Acquisition of data, Analysis and interpretation of data; MMUA, Conception and design, Analysis and interpretation of data, Drafting or revising the article

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
