## [Decision Letter]

Thank you for sending your work entitled “Unconventional binding of BiP ATPase to Ire1 and Perk is dissociated by unfolded proteins to initiate stress signaling” for consideration at *eLife*. Your article has been favorably evaluated by Randy Schekman (Senior editor) and three reviewers, one of whom is a member of our Board of Reviewing Editors.

The Reviewing editor and the other reviewers discussed their comments before we reached this decision, and the Reviewing editor has assembled the following comments to help you prepare a revised submission.

The ER-localized Hsp70 chaperone BiP/Grp78 has been shown to associate with the ER stress sensors, Ire1 and Perk, under unstressed conditions. ER stress dissociates BiP from both sensors, triggering dimer/oligomer formation and activation of the cytosolic effector domains of the sensors by trans-autophosphorylation. Dissociation of BiP is thus required to activate IRE1 and/or Perk. In yeast, which have an evolutionarily conserved Ire1, subregion V, juxtaposed to transmembrane domain of Ire1, has been identified as a BiP-binding region (Ref. 13; PLoS Biol. 8, e1000415, 2010), and the BiP binding region of mammalian Perk has also been identified as the membrane proximal, luminal segment (JBC, 277, 18728, 2002). Two groups have reported that the luminal core domain of yeast Ire1 interacts directly with unfolded or misfolded protein in vivo and in vitro (Ref. 14; MBC, 22, 3520, 2011; PNAS, 102, 18773, 2005; Science, 333, 1891, 2011).

The paper by Ali et al. presents data showing that the luminal domains of Ire1 and Perk (between 40 and 50 kD each) bind BiP, but not through subregion V. The ATPase domain of BiP, not the substrate domain, is the site of interaction, which is nonetheless independent of nucleotide. Addition of an unfolded protein -- the C_H_1 region of an immunoglobulin -- to full-length BiP eliminates the observed association with the luminal domain of either Ire1 or Perk. Unfolded C_H_1 does not bind the luminal domains themselves, leading to the conclusion that the conformational change in BiP engendered by peptide binding must in some way alter access to the site(s) of luminal-domain binding. The authors have used microscale thermophoresis to determine luminal-domain binding and confirmed the results by pull-down, which has then been used to study the effect of C_H_1 binding. The conclusion, assuming it were to hold up in cells, would be biologically satisfying, with the implication that BiP inhibits Perk and Ire1 signaling unless bound with an unfolded protein. BiP would then be the primary unfolded protein detector, consistent with its known activities, and Perk and Ire1 would be downstream signal transducers.

The reviewers concurred that the results are interesting and that they constructively challenge existing models. The reviewers also concurred that this revisionist view will not be convincing without in vivo data. The authors therefore need to carry out experiments that address the following points:

1) Does the proposed model hold in vivo? One way to ask this question is to show that BiP mutants deficient in substrate binding retain Ire1/Perk inhibition and hence attenuate Ire1 and Perk activation, even in the presence of unfolded protein (ER stress). Readout would be the state of Ire1 and Perk phosphorylation, XBP1 mRNA splicing, or other indicators of the activated state. The authors may have a better way. In any case, they need to include clear predictions that they test in cells.

2) Does deletion or alteration of subregion V in Ire1 and Perk affect the binding of their luminal regions to BiP in cells? The authors need to test their model, which differs from published data.

3) An in vitro experiment: Luminal domains of both Ire1 and Perk are known to be dimers and to oligomerize. Apart from their supplementary data showing that these domains are dimers and unaffected by the EspP signal peptide, he authors of the present study do not directly address the state of oligomerization of the luminal domains. Is it affected by BiP binding?

4) Discussion points:

a) Cite and discuss some of the past work and how it differs or is consistent with the present results, and admit some of the complexities that may impact the interpretation of the present results. For example, a previous study by Pincus et al. in PLoS Biology in 2010 looked at the interactions between BiP and the Ire1 luminal domain and offered a model. How do the two studies differ? How can one reconcile them?

b) Table 1 shows that human Ire1a lacks subregion 1 (only 8 amino acids). Since the data on Ire1(32aa-440aa) are mostly similar to those of Ire1(24aa-440aa), the latter are enough, and the interpretation that human Ire1a comprises only subregions II-V regions seems to be simple and correct. The current presentation seems to imply, incorrectly, that human Ire1 has a long subregion 1, like yeast.

The editor will be happy to consider a resubmission that contains new data responding to points 1-3 and that expands the Discussion in line with point four.

[Editors' note: further revisions were requested prior to acceptance, as described below.]

Thank you for resubmitting your work entitled “Noncanonical binding of BiP ATPase domain to Ire1 and Perk is dissociated by unfolded protein C_H_1 to initiate ER stress” for further consideration at *eLife*. Your revised article has been favorably evaluated by Randy Schekman (Senior editor) and a member of the Board of Reviewing Editors. The manuscript has been improved but there are some remaining issues that need to be addressed before acceptance, as outlined below.

Many groups have studied the association of BiP with Ire1 and Perk in vivo by IP. In yeast Ire1, BiP binding region has been assigned to a subregion V, a region neighboring transmembrane domain of Ire1 by Kimata et al. (2004, J Cell Biol, 167, 445-456; Promlek, 2011, Mol Biol Cell 22, 3520-3532) and another group provided supporting evidence (Ref. 20). Subregion V of mammalian Ire1 has been also reported as a BiP binding region, but this region is longer than the corresponding region in yeast Ire1 (Oikawa et al., ECR, 2009). The deletion of subregion V greatly decreases the basal level of BiP binding, and the deletion of both subregion 1 and 5 increases basal activation level of yeast Ire1 (Kimata et al., 2004, J Cell Biol, 167, 445-456; Ishiwata et al., Genes Cells, 2013). In mammalian Ire1, the deletion of subregion V decreased the amount of Ire1 but increased the basal Ire1 activity (Oikawa et al., ECR, 2009). Subregion V of Perk is also reported as a BiP binding region ([11], J Biol Chem 277, 18728-18735). These data clearly show that the deletion of subregion V of Ire1 and Perk decreases BiP binding and increases Ire1 basal activity, which is tightly regulated. Furthermore Dr. Walter's group reported that the groove constructed by a dimer of the Ire1 core region directly binds unfolded proteins ([4], PNAS, 102, 18773-18784), an idea supported by other reports (Kimata et al., The Journal of Cell Biology, 179, 75-86; Gardner et al., 2011, Science 333, 1891-1894; Promlek, 2011, Mol Biol Cell 22, 3520-3532).

In vivo experiments are especially important for clarifying the authors’ claim, and the new experimental data in the revised manuscript are not sufficient.

The authors should carry out the following experiments:

1) They should show the in vivo interactions of the BiP ATPase domain with Ire1 and Ire1ΔV, respectively, and compare them.

2) In Figure 3, their data show only an interaction between BiP and Ire1ΔV; the authors should compare Ire1ΔV-BiP association with that of wild-type Ire1-BiP. Because BiP is an ER chaperone that binds many proteins, it is not surprising that BiP interacted with Ire1ΔV in their experiments. They need to coexpress full length Ire1 and Ire1ΔV (separately, of course) with full length BiP in the absence or presence of ER stress. If their claim were correct, the data would show the following:

a) In the absence of ER stress, similar amounts of BiP should associate with Ire1 and Ire1ΔV.

b) In the presence of ER stress, the amount of BiP associated with Ire1 should decrease, and the decreased level should not depend on the deletion of region V.

3) In Figure 6, the authors should examine not only Ire1 and Perk but also Ire1ΔV and PerkΔV.

---

## [Author Response]

*1) Does the proposed model hold in vivo? One way to ask this question is to show that BiP mutants deficient in substrate binding retain Ire1/Perk inhibition and hence attenuate Ire1 and Perk activation, even in the presence of unfolded protein (ER stress). Readout would be the state of Ire1 and Perk phosphorylation, XBP1 mRNA splicing, or other indicators of the activated state. The authors may have a better way. In any case, they need to include clear predictions that they test in cells*.

To address this question, we have used a BiP deletion mutant that lacks the substrate binding domain (encompassing only ATPase domain), consistent with constructs used for the in vitro analysis. We have, as the reviewers have suggested, used the state of Ire1 and Perk phosphorylation as readouts of activated state. In line with our predictions, the BiP deletion mutant, lacking the substrate binding domain, displays attenuated UPR signaling in comparison to full length BiP. This result greatly enhances the manuscript. See section “BiP deletion mutants that lack substrate binding domain attenuate UPR signaling” and Figure 6.

*2) Does deletion or alteration of subregion V in Ire1 and Perk affect the binding of their luminal regions to BiP in cells? The authors need to test their model, which differs from published data*.

We conducted a co immunoprecipitation experiment using Ire1 mutant that lacked region V (Ire1 ΔV) with HA-tagged BiP. We can confirm that we see Ire1 ΔV co-immunoprecipitating with BiP, reinforcing the in vitro data, where we measured robust binding (kd = ∼2μM) between Ire1 ΔV and BiP. This data clearly suggests that region V is dispensable for BiP binding. See section “BiP binds to region II-IV of luminal domains” and Figure 3.

3) An in vitro experiment: Luminal domains of both Ire1 and Perk are known to be dimers and to oligomerize. Apart from their supplementary data showing that these domains are dimers and unaffected by the EspP signal peptide, he authors of the present study do not directly address the state of oligomerization of the luminal domains. Is it affected by BiP binding?

We decided to use cross-linking as a method to address this question. The difficulty in this experiment is obtaining a result that is clear to interpret, and key to this experiment working was using the ATPase domain of BiP. The ATPase domain interacts with Ire1 exclusively, without the requirement of substrate binding domain. But more importantly, it remained a monomer in presence and absence of cross linker. Thus, making analysis of multimer bands an easier task. The cross link clearly identifies Ire1 as a dimer and tetramer species. Upon addition of BiP, and subsequent immunoblotting, reveals that BiP binds to Ire1 in two possible ratios. The key conclusion here is that the presence of BiP impedes the formation of tetramers by binding to Ire1 monomers and dimers. See section “BiP impedes Ire1 luminal domain tetramer formation” and Figure 7.

4) Discussion points:

*a) Cite and discuss some of the past work and how it differs or is consistent with the present results, and admit some of the complexities that may impact the interpretation of the present results. For example, a previous study by Pincus et al*. *in PLoS Biology in 2010 looked at the interactions between BiP and the Ire1 luminal domain and offered a model. How do the two studies differ? How can one reconcile them?*

After being encouraged to discuss previous work within the field, we have given an open and honest opinion of the previous models proposed, and where we think some weaknesses may lie. Our viewpoint is that from in vitro analysis experts, whilst the majority of the field comes from the in vivo analysis side. Our model suggests an allosteric mechanism to UPR induction.

*b)*
Table 1
*shows that human Ire1a lacks subregion 1 (only 8 amino acids). Since the data on Ire1(32aa-440aa) are mostly similar to those of Ire1(24aa-440aa), the latter are enough, and the interpretation that human Ire1a comprises only subregions II-V regions seems to be simple and correct. The current presentation seems to imply, incorrectly, that human Ire1 has a long subregion 1, like yeast*.

We have addressed this point by inputting the statement below into the section

“BiP binds to region II-IV of luminal domains” and have edited Figure 3 to show that region I is very small”

“One point to note, is that using this assignment (Kimata, Y. 2004; Kimata, Y 2007) yeast Ire1 luminal domain possesses an extended region I, whilst the equivalent region in human Ire1 is essentially absent. The implication for human Ire1 is that both regions I and II are very close together and map onto the equivalent of yeast domain II.”

[Editors' note: further revisions were requested prior to acceptance, as described below.]

We have now complied with the reviewers’ comments, points 1 and 2. The inclusion of the new data has greatly improved the manuscript.

*The authors should carry out the following experiments*:

*1) They should show the in vivo interactions of the BiP ATPase domain with Ire1 and Ire1ΔV, respectively, and compare them*.

We have now done a co-immunoprecipitation experiment showing an interaction between BiP ATPase with Ire1 and Ire1 ΔV, this is presented in Figure 3 and in the third paragraph of the Results section of manuscript. This interaction confirms the previous in vitro interaction measured by thermophoresis at ∼2μM between BiP ATPase and luminal domains (both region I-V and II-IV, Figure 3).

*2) In*
Figure 3*, their data show only an interaction between BiP and Ire1ΔV; the authors should compare Ire1ΔV-BiP association with that of wild-type Ire1-BiP. Because BiP is an ER chaperone that binds many proteins, it is not surprising that BiP interacted with Ire1ΔV in their experiments. They need to coexpress full length Ire1 and Ire1ΔV (separately, of course) with full length BiP in the absence or presence of ER stress*.

The reviewers correctly pointed out that our first attempt (first revision) to measure BiP and Ire1 ΔV was deficient; we needed to repeat the experiment along side FL Ire1, in the absence and presence of ER Stress. We now have repeated our experiment with the controls mentioned by the reviewers (Figure 3 and third paragraph of the Results section). We indeed see an interaction with Ire1ΔV, and when cells are challenged with tunicamycin to induce ER stress, we see a reduction in both FL and ΔV Ire1 interacting with BiP that is in not dependent upon construct size. This confirms the previous in vitro analysis that measures an interaction with Ire1ΔV at ∼2 μM in Figure 3.